# Identification of Late Ripening Citrus Mutant, *Ara-unshiu* (*Citrus unshiu*), and Its Selectable Marker

**DOI:** 10.3390/plants12193355

**Published:** 2023-09-22

**Authors:** Ji-Man Heo, Chang-Ho Eun, In-Jung Kim

**Affiliations:** 1Department of Biomaterials Engineering, Graduate School, Jeju National University, Jeju 63243, Republic of Korea; cgi24@jejunu.ac.kr; 2Subtropical Horticulture Research Institute, Jeju National University, Jeju 63243, Republic of Korea; mong6908@gmail.com; 3Bio-Resources Computing Research Center, Research Institute for Subtropical Agriculture and Biotechnology, SARI, Jeju National University, Jeju 63243, Republic of Korea

**Keywords:** *Citrus unshiu*, SNP, InDel, late ripening, selection marker, AS-PCR

## Abstract

‘Miyagawa-wase’ mandarin (*Citrus unshiu* Marc. cv. *Miyagawa-wase* early) is one of the most widely cultivated citrus varieties on Jeju Island in Korea. Mutation breeding is a useful tool for the induction of genetic diversity for the rapid creation of new plant variants. We previously reported the use of gamma irradiation for the development of new citrus varieties. Here, we report a new mutant, *Ara-unshiu*, with a unique late fruit ripening phenotype. We investigated the fruit morphological characteristics including weight, vertical/transverse diameter, peel thickness, hardness, and color difference, as well as sugar and acid contents of the *Ara-unshiu* compared to wild-type controls. We then used whole genome re-sequencing and functional annotation by gene ontology to identify and characterize single nucleotide polymorphism (SNP) and insertion/deletion (InDel) variants in the *Ara-unshiu*, finding a greater abundance of annotated genes containing InDels compared to SNPs. Finally, we used allele-specific PCR to identify molecular markers among the homozygous SNPs detected from the *Ara-unshiu* genome sequencing. We report a primer set that effectively distinguishes the *Ara-unshiu* from the wild-type control and other citrus varieties. Our findings provide insights into the mechanisms controlling the timing of fruit ripening and tools for the molecular breeding of citrus varieties.

## 1. Introduction

Citrus is a fruit crop genus with significant worldwide economic importance. Among citrus fruit, ‘*Miyagawa-wase*’ mandarin (*Citrus unshiu* Marc. cv. *Miyagawa-wase*) has become one of the most widely cultivated varieties, accounting for more than 80% of the total citrus cultivation on Jeju Island in Korea. According to the 2023 Korea National Statistical Office report, Jeju Island’s citrus cultivation area is the largest at 15,818 ha, followed by plums and persimmons at 56 ha and 55 ha, respectively. The improvement of citrus varieties through conventional breeding is difficult due to several reproductive biological features, such as apomixis, male and female partial sterility, self- and cross-incompatibility, a long juvenile period, and a high level of heterozygosity [1,2,3]. Improvement of the *Miyagawa-wase* cultivar by crossbreeding is further complicated by its seedless phenotype. Although this cultivar can be crossed with a seed-forming maternal citrus parent, one of the major problems that ensues is polyembryony, by which one seed may contain two or more embryos with zygotic or nucellar origin [4]. Thus, these zygotic embryos are usually stunted and exhibit poor growth. Transgenic plant approaches can also be used to develop superior citrus varieties, but restrictions placed on genetically modified organisms make them difficult to distribute across global markets [5].

An alternative approach is mutation breeding, where mutagenesis using irradiation or chemical mutagens is used to induce genetic diversity for creating new varieties in a short period of time [6]. Mutation breeding approaches overcome the limitation imposed by the extremely low frequency of spontaneous mutations and are, therefore, increasingly considered a powerful tool for plant breeding [7]. Of these approaches, the wide availability and penetrance of gamma irradiation have made it a central approach used in fruit tree species such as grapefruit, pomelo, lemon, and mandarin [8,9,10,11,12]. As a physical mutagen, gamma rays react with molecules or atoms within cells to form free radicals, which induce a higher incidence of genetic mutations by damaging chromosomal DNA [13]. The chosen irradiation dose has a variable influence on the morphology and physiology of the plants, ranging from low-dose stimulation to high-dose inhibition [14].

We recently reported a new mutant, *Jedae-unshiu*, induced by gamma irradiation of *C. unshiu* Marc. cv. *Miyagawa-wase* scions. *Jedae-unshiu* has a unique fruit shape with vertical troughs on the flavedo, smooth albedo without rough protruding fibers, and good adhesion between peel and flesh [15]. In the present study, we uncovered another novel *C. unshiu* mutant, *Ara-unshiu*, which shows a fruit phenotype with delayed ripening compared to the wild-type (WT) control fruit. Fruit ripening, the final stage of fruit development, is a complex process involving a series of physiological and biochemical changes. The ripening of fruits is always accompanied by a rapid change in color as pigments accumulate and chlorophyll breaks down. Simultaneously, the fruit softens through cell wall remodeling, and flavor develops as acids, sugars, and volatile compounds accumulate. Fruit ripening programs are known to be mediated by the combined effects of plant hormones, transcription factors, and epigenetic modifications, all of which can affect fruit quality [16,17,18]. Several citrus ripening mutants have been reported to date. The Fengjiewancheng mutant (*Citrus sinensis* L. Osbeck) derived from a bud mutation of the Fengjie 72-1 navel orange ripens one month later than its parental line [19]. Fengwan and Jincheng mutants from *Citrus sinensis* have also been reported as spontaneous late-ripening mutants [20,21]. *Tardivo*, another late-ripening mutant of the Clementine mandarin, is associated with an altered perception and sensitivity to ethylene [22]. Conversely, the *Liuyuezaoyou* mutant is an early-ripening cultivar selected from a bud mutation of Guanximiyou (*Citrus grandis* Osbeck) [23]. The Ganqi 4 mutant was also reported as a spontaneous early-ripening navel orange mutant of Newhall (*Citrus sinensis* L. Osbeck) [24]. Tomato, strawberry, and pomelo mutants with altered timing of fruit ripening were also recently reported [23,25,26,27]. To further characterize the process of fruit ripening, we investigated the morphological traits and sugar/acid contents of the late-ripening mutant *Ara-unshiu*. In addition, we conducted whole genome re-sequencing to identify specific molecular markers for *Ara-unshiu* mutant selection. Our results contribute to the understanding of the process of fruit ripening and provide a foundation for revealing the molecular mechanisms of fruit maturation.

## 2. Materials and Methods

### 2.1. Plant Materials

The citrus mutant *Ara-unshiu* was produced by gamma irradiation of wild type (WT) *C. unshiu* Marc. cv. *Miyagawa-wase*. These materials were planted in the Research Institute for Subtropical Agriculture and Biotechnology of Jeju National University, Seogwipo, Republic of Korea. Young leaves and immature green fruits (late June fruit) were harvested and immediately frozen in liquid nitrogen for genome re-sequencing analysis. Other citrus varieties used in this study were as follows: *Jedae-unshiu* (J), Chung-Gyeon (CG), Gam-Pyong (GP), Hanla-Bong (HB), and Kara-Hyang (KH). *Jedae-unshiu* has an early maturity and the others have a late maturity trait.

### 2.2. Analysis of Fruit Traits

Fruit trait analysis was performed according to the method of Eun and Kim [15]. Fruit weight was measured using an electronic indicator scale (CAS Co., Ltd., Yangju, Republic of Korea). The vertical diameter, transverse diameter, and peel thickness of the fruit were measured using a digital caliper (MITUTOYO Corporation, Kawasaki, Japan). Fruit hardness was measured using a fruit hardness meter (LUTRON FR-5105, Antala Staška, Czech Republic). Sugar content and acidity were measured in 4–5 mL of fruit juice according to the instruction manual for the NH-2000 (HORIBA, Kyoto, Japan). Changes in the fruit peel color were measured using a chromometer (CR-400, MINOLTA, Tokyo, Japan). For both the *Ara-unshiu* and WT control, 10 or more fruits per tree from five or more trees each year, over the course of two years, were harvested and investigated.

### 2.3. Whole Genome Re-Sequencing, Detection of Single Nucleotide Polymorphism (SNP) and Insertion/Deletion (InDel) Variants, and Gene Ontology (GO) Analysis

Genomic DNA from leaves of the *Ara-unshiu* and WT control was extracted using a cetyltrimethylammonium bromide (CTAB) method [28] and sequenced on the Illumina HiSeq X Ten platform by Macrogen company (Daejeon, Republic of Korea). Sequence data pre-processing, alignment to the reference genome, SNP and InDel detection and classification, and GO analysis were performed as described by Eun and Kim [29]. The reference genome information for *C. unshiu* Marc. *Miyagawa-wase* was obtained from the NCBI (assembly CUMW_v1.0) and the GenBank assembly (accession: GCA_002897195.1).

### 2.4. Allele-Specific PCR Markers

To identify specific selection markers for the *Ara-unshiu* mutant, allele-specific PCR (AS-PCR) primers were designed according to Bui and Lit [30]. Five homo-type SNPs were selected among variants between the *Ara-unshiu* and WT control. For the allele-specific primer containing the target SNP allele of *Ara-unshiu*, an additional nucleotide mismatch was introduced at the second base position upstream of the 3′ terminus. The other primer employed the same sequence for both the *Ara-unshiu* and the WT control (Appendix A). For the PCR amplification control, the same forward and reverse primers were used to amplify PCR products of both the *Ara-unshiu* and WT control. PCR was performed in a final volume of 20 µL using TOPsimple DryMIX-nTaq (Enzynomics, Seoul, Republic of Korea), 5 µM of each primer, and 50 ng of DNA. PCR conditions were as follows: an initial denaturing at 95 °C for 3 min; 30 cycles at 95 °C for 30 s, 55 °C for 30 s, and 72 °C for 20 s; and a final extension at 72 °C for 5 min. PCR products were electrophoresed in 1.5% (*w*/*v*) agarose gels. Besides DNA samples from the *Ara-unshiu* and WT control, DNA samples from other citrus varieties were examined, including Chung-Gyeon (CG, *C. kiyomi*), Gam-Pyong (GP, *C. hybrid* cv. Kanpei), Hanla-Bong (HB, *C. reticulata* Shiranui), and Kara-Hyang (KH, *C. hybrid* Natsumi).

## 3. Results and Discussion

### 3.1. Selection of Mutant Lines by Gamma Irradiation

To induce mutant lines of *C. unshiu* (Marc. cv. Miyagawa-wase), scions were irradiated with gamma rays (^60^Co, 100Gy with 60% survival rate of budsticks) followed by grafting on branches of Miyagawa-wase (*C. unshiu* Marc.) [31]. Previous studies of mutant fruits induced by gamma irradiation mainly focused on the enhancement of the functional components of fruit, their storage properties, or on the induction of seedless fruits. In contrast, in this study, we selected mutants showing late fruit ripening and identified a mutant line that we named *Ara-unshiu*. The following year, *Ara-unshiu* scions were used for veneer grafting on the rootstocks (*C. unshiu* Marc. cv. *Miyagawa-wase*) and continuously monitored for the maintenance of the late-ripening trait [31,32].

### 3.2. Comparison of Morphological Traits between Ara-unshiu and Wild Type (WT) Fruit

We first investigated the external fruit morphological differences between WT control and *Ara-unshiu*. At the mature fruit stage (late November), control fruit were fully orange-colored; however, *Ara-unshiu* fruit were still green-colored on the peel (Figure 1). In addition, the *Ara-unshiu* exhibited better adhesion between peel and flesh than the control fruit. Across two years of study (2021 to 2022), *Ara-unshiu* fruit were similar to WT in horizontal length, vertical length, fruit weight, and peel thickness, but had higher fruit hardness compared to the WT control (Table 1). Comparison of Hunter color values between the *Ara-unshiu* and WT control peels showed that the red value (a) of the *Ara-unshiu* (13.76 ± 2.52 in 2021 and 6.84 ± 8.89 in 2022) was lower than the WT control (25.22 ± 0.72 in 2021 and 25.92 ± 2.01 in 2022); however, no significant differences in white (L) or yellow (b) values were noted. Sugar levels and acidity also showed similar values between *Ara-unshiu* and WT control fruit (Table 2).

### 3.3. Mapping of Re-Sequencing Reads to C. unshiu Marc. Miyagawa-wase CUMW_v1.0

We recently reported that the citrus mutant *Jedae-unshiu*, induced by gamma irradiation, shows a unique fruit phenotype with vertical troughs on the flavedo and smooth albedo without rough protruding fibers [15]. To further understand the ripening delay phenotype of the *Ara-unshiu* mutant, whole genome sequencing of *Ara-unshiu* was conducted to identify DNA sequence polymorphisms between *Ara-unshiu* and our previously sequenced WT control (NCBI: PRJNA745525) [29]. *C. unshiu* Marc. *Miyagawa-wase* (CUMW_v1.0) was used as a reference genome and had a length of 359.7 megabases (Mb) [33]. After pre-processing the raw sequence data from the *Ara-unshiu* and WT control, we obtained 80,976,944 and 80,693,250 of clean reads, respectively. The rate of mapped reads was 86.77% and 86.75% and the average coverage per sample was 29.50× and 27.49× for *Ara-unshiu* and WT, respectively (Table 3).

Compared with the reference genome, 1,198,650 and 1,193,106 SNPs were present in the WT and *Ara-unshiu*, respectively (Appendix A). Of those, there were 8208 and 9457 homozygous SNPs and 572,811 and 576,678 heterozygous SNPs in the WT and *Ara-unshiu*, respectively. We then classified the genome annotations of the SNPs in the WT and *Ara-unshiu* (Appendix A). In the WT, 367,591 SNPs were detected in genic regions (164,743 in exons and 212,665 in introns) and 715,536 were detected in intergenic regions. In the *Ara-unshiu*, 366,568 SNPs were detected in genic regions (164,072 in exons and 212,363 in introns) and 712,344 were detected in intergenic regions. We also detected InDels in the WT and *Ara-unshiu* genome sequences after comparison with the reference genome (Appendix A). In the WT, 172,259 InDels were detected, of which homozygous and heterozygous InDels constituted 3751 (1600 insertions and 2151 deletions) and 57,930 (29,136 insertions and 28,794 deletions) events, respectively. In the *Ara-unshiu*, 172,154 InDels were detected, of which homozygous and heterozygous InDels constituted 3877 (1648 insertions and 2229 deletions) and 57,351 (28,708 insertions and 28,643 deletions) events, respectively. Classification by genome annotation (Appendix A) showed that of the 155,928 InDels in the WT, 45,362 were classified as genic (3799 in coding sequences [CDSs], 11,128 in exons, and 35,441 in introns) and 110,566 were detected in intergenic regions. In the *Ara-unshiu*, the 155,751 InDels were classified as 45,353 genic (3757 in CDSs, 11,105 in exons, and 35,448 in introns) and 110,398 intergenic. Thus, the patterns of SNP and InDel detection and classification by genome annotation were similar between the two samples, with both SNPs and InDels mainly located in intergenic regions. The detection of a large number of SNPs and InDels in both the WT and *Ara-unshiu* compared to the reference genome indicates the published reference genome sequence is quite different from our WT sequence despite the two belonging to the same cultivar. Similarly, Eun and Kim [29] reported that *C. unshiu* Marc. cv. *Miyagawa-wase* plants cultivated in different geographical region shows genome-wide DNA sequence variation including SNPs and InDels, and Torkamaneh et al. [34] reported that the alignment of two reference genome sequences from two different regions provides evidence for the existence of structural variants between individual samples. Taken together, we suggest that genome re-sequencing of a control sample should be considered in genome-wide comparative studies.

### 3.4. SNP and InDel Genetic Variation between WT and Ara-unshiu

To identify the mutation(s) responsible for the late ripening of the *Ara-unshiu*, we compared the genomic sequences of the WT and *Ara-unshiu* and uncovered 3469 SNPs (46 homozygous and 3423 heterozygous) and 7162 InDels (45 homozygous and 7117 heterozygous) (Table 4). Of these, we further analyzed SNP and InDel variation in coding regions. A total of 2057 SNPs and 3612 InDels were detected in the CDSs of 364 and 1108 genes annotated in the GO database, respectively (Table 5). Thus, by direct comparison of genetic variation between the two samples, we eliminated many of the non-specific SNPs and InDels. These results indicated that the *Ara-unshiu* genome sequence mutations included more InDels than SNPs. This is similar to the genome re-sequencing of the *Jedae-unshiu* mutant, where polymorphic InDels were more numerous than SNPs [15].

### 3.5. Functional Annotation of Genes Containing SNPs and InDels in Ara-unshiu

Non-synonymous SNPs and InDels that occur in CDSs could alter gene function. Thus, we identified genes that contained SNPs and InDels in the *Ara-unshiu* compared to the WT control that were annotated in the GO database and performed GO analysis on them. GO analysis classified the variant genes by biological process (BP), cellular component (CC), and molecular function (MF) categories (Figure 2). SNP variants within 874 genes were classified into 115 BP subcategories, with response to stress (96), cellular response to stress (42), response to inorganic substance (40), oxidation-reduction process (32), and response to metal ion (20) being the main subcategories. With regard to the MF, 831 genes were classified into 55 subcategories, including binding (176), catalytic activity (166), small molecule binding (68), hydrolase activity (62), and catalytic activity (acting on a protein) (58) (Figure 2A). None of the genes were classified into any significantly enriched CC subcategories. For InDels, 3019 genes were classified into 184 BP subcategories, with cellular process (794), negative regulation of biological process (113), transmembrane transport (91), cell cycle (72), and organic hydroxy compound metabolic process (71) being the main subcategories. With regard to CC, 427 genes were classified into 18 subcategories including protein-containing complex (121), intracellular non-membrane-bounded organelle (85), catalytic complex (53), transferase complex (43), and intracellular protein-containing complex (38). For MF, 3742 genes were annotated in 149 subcategories, including binding (575), organic cyclic compound binding (335), heterocyclic compound binding (333), small molecule binding (203), and nucleoside phosphate binding (185) (Figure 2B). These results indicated that InDel variants were more often annotated by GO than SNP variants were. Furthermore, the top five BP subcategories were annotated differently between genes containing SNPs and InDels, but some of the subcategories of MF were identical. Considering the late-ripening fruit trait of the *Ara-unshiu*, the genes included in the BP annotations are more promising candidates compared to genes included in the CC or MF annotations.

### 3.6. Identification of Specific Selection Markers for Ara-unshiu

AS-PCR is a method for SNP detection based on the extension of a PCR primer only when its 3′ end is perfectly complementary to the template [35,36]. AS-PCR is simple, rapid, economical, reliable, and does not require any sophisticated technology or instrumentation. Therefore, it has been widely used to screen for genetic markers among target SNP genotypes from a large number of natural populations [37,38] or mutant segregant populations [39,40,41]. To identify specific selection markers for the *Ara-unshiu* using AS-PCR, five homo-type SNPs (Ara-SNP1 through Ara-SNP5) were selected from the SNP variants between the *Ara-unshiu* and WT control. We then designed a pair of common primers and a specific primer with an introduced mismatch in the vicinity of each SNP site (Appendix A). Of the five sets of primers, Ara-SNP4 was effectively able to distinguish the *Ara-unshiu* from WT control and other citrus varieties (J, CG, GP, HB, and KH) (Figure 3). The SNP site of Ara-SNP4 marker was located on the intergenic region of the genome. This established the Ara-SNP4 primer set as a specific selection marker for the *Ara-unshiu*. The position of the introduced mismatched base is critical for the success of AS-PCR. The penultimate (−2) or antepenultimate (−3) position of the 3′ end of the mismatched primer are generally used to introduce the mismatched base to maximize the specificity of detection and avoid false positives [30,35]. Accordingly, we introduced the mismatched base at the −2 position of the 3′ end of the mismatched primer. The type of mismatched base also affects the specificity of the primers; the specific mismatched primers in this study were designed according to Bui and Liu [30].

In summary, we report a new citrus mutant induced by gamma irradiation, *Ara-unshiu*, that carries a late fruit ripening phenotype. We report the comparison of several fruit morphological traits as well as sugar/acid contents between the *Ara-unshiu* and its WT control. We then used AS-PCR to identify a specific selection marker (Ara-SNP4) that distinguishes the *Ara-unshiu* from other citrus varieties.

## Figures and Tables

**Figure 1 plants-12-03355-f001:**
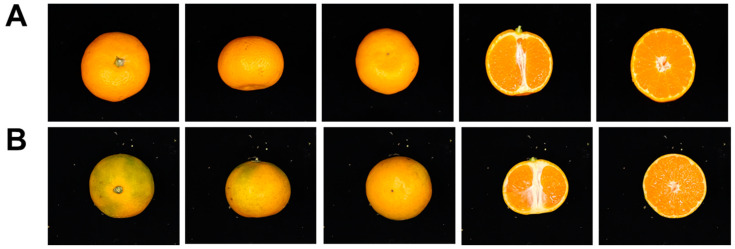
Morphological comparison of fruit between wild type (WT) and *Ara-unshiu*. From left to right: top view, side view, bottom view, longitudinal section, transverse section. (**A**) Wild type. (**B**) *Ara-unshiu.*

**Figure 2 plants-12-03355-f002:**
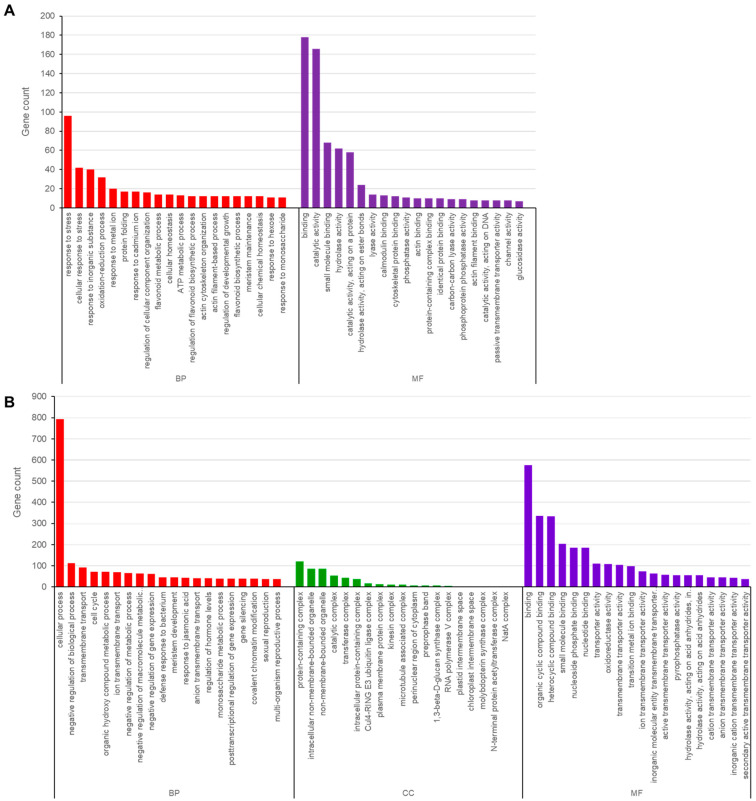
Gene ontology (GO) functional enrichment of genes containing single nucleotide polymorphism (SNP) and insertion/deletion (InDel) variants between wild type and *Ara-unshiu*. (**A**) The ten most enriched subcategories of biological process (BP) and molecular function (MF) among genes containing SNP variants. (**B**) The ten most enriched subcategories of BP, cellular component (CC), and MF among genes containing InDel variants.

**Figure 3 plants-12-03355-f003:**
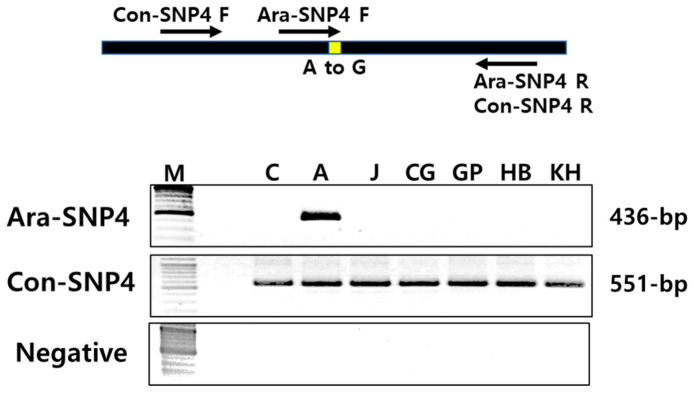
Identification of an *Ara-unshiu* selection marker by allele-specific PCR. C: WT control, A: *Ara-unshiu*, J: *Jedae-unshiu*, CG: Chung-Gyeon, GP: Gam-Pyong, HB: Hanla-Bong, KH: Kara-Hyang. Negative: all PCR components without template DNA. The yellow and the arrows show a SNP site and primer direction, respectively.

**Table 1 plants-12-03355-t001:** Comparative analysis of wild type (WT) control and *Ara-unshiu* fruit.

	Year	Horizontal Length	Vertical Length	Single Fruit Weight	Peel Thickness	Hardness
		(mm)	(mm)	(g)	(mm)	(G)
WT	2021	46.27 ± 3.89	58.45 ± 4.03	87.60 ± 13.73	2.12 ± 0.22	937 ± 58
	2022	47.83 ± 3.68	62.32 ± 6.64	98.94 ± 29.29	2.65 ± 0.43	862 ± 119
*Ara-unshiu*	2021	50.78 ± 2.39	58.82 ± 3.18	95.41 ± 15.19	2.27 ± 0.55	1352 ± 86
	2022	53.06 ± 8.76	60.28 ± 9.96	108.9 ± 30.60	2.28 ± 0.76	1145 ± 444

**Table 2 plants-12-03355-t002:** Comparison of sugar level, acidity, and Hunter color values between WT control and *Ara-unshiu* fruit.

	Year	Hunter Color Values			Total Soluble Solid	Acidity
		L (Light)	a (Red)	b (Yellow)	(Brix)	(wt%)
WT	2021	59.36 ± 1.50	25.22 ± 0.72	35.47 ± 0.59	9.38 ± 0.27	0.72 ± 0.08
	2022	59.47 ± 1.62	25.92 ± 2.01	35.28 ± 1.05	9.41 ± 0.32	0.46 ± 0.03
*Ara-unshiu*	2021	63.04 ± 3.33	13.76 ± 2.52	37.21 ± 2.58	8.96 ± 0.27	0.78 ± 0.07
	2022	62.94 ± 9.52	6.84 ± 8.89	37.17 ± 5.85	9.27 ± 1.48	0.73 ± 014

**Table 3 plants-12-03355-t003:** Statistics for re-sequencing data in WT and *Ara-unshiu.*

Sample	Clean Reads ^1^	Mapped Reads ^2^	Mapped Regions ^3^ (%)	Coverage ^4^
WT	80,693,250	71,296,288 (88.35%)	310,983,030 (86.75%)	≒27.49×
*Ara-unshiu*	80,976,944	77,318,919 (88.90%)	312,053,993 (86.77%)	≒29.50×

^1^ The number of clean reads that passed pre-processing and were used for read alignment. ^2^ The number of clean reads that mapped to the reference genome sequence when aligned. ^3^ The total length (in base pairs) of aligned reads. Percentage is with regard to the reference genome. ^4^ The value obtained by dividing the total read length of each sample by the assembled genome size (359.7 Mb).

**Table 4 plants-12-03355-t004:** Quantification of SNPs and InDels between WT and *Ara-unshiu.*

		Number ^1^	Homozygous	Heterozygous
WT vs. *Ara-unshiu*	SNP	3469	46	3423
	InDel	7162	45	7117

^1^ SNP and InDel loci showing differences in nucleotide sequences between comparative samples.

**Table 5 plants-12-03355-t005:** Statistics of genes containing SNPs and InDels between WT and *Ara-unshiu.*

		Number	Within Genes ^1^	GO Genes ^2^
WT vs. *Ara-unshiu*	SNP	2057	323	364
	InDel	3612	1180	1108

^1^ Number of genes containing SNPs of InDels. ^2^ Number of genes containing SNPs or InDels and belonging to enriched GO annotations (e-value ≤ 1 × 10^−10^, best hits).

## Data Availability

Original sequence data can be found in the NCBI Sequence Read Archive with the following accession numbers: PRJNA745525 for WT (Miyagawa-1) and PRJNA996743 for *Ara-unshiu*. Other data are available upon request.

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
