# Peer review of "Identification of Late Ripening Citrus Mutant, Ara-unshiu (Citrus unshiu), and Its Selectable Marker"

_plants, 2023, doi:10.3390/plants12193355_

Round 1
Reviewer 1 Report
The authors obtained a late-maturing citrus mutant and identified its differences from the parent through comparative analysis of multiple indicators. A large number of different sites were obtained through whole genome sequencing mining, which provides useful sequence information for the breeding of this citrus. However, there are some problems in the paper, which need to be revised
1. In the first paragraph of the results section, the author wrote that the mutant was obtained by gamma irradiation treatment, but the treatment method and intensity were not mentioned in the materials and methods section.
2. The Figure 1 lacks the necessary scale, which mistakenly leads people to believe that the mutant is significantly smaller than WT, but the result is not as can be seen from Table 1.
3. What do the letters of a b L stand for in Table 2? Lack the necessary information.
4. The information of Table S5 is missing in the online document.
5. Citrus varieties (CG, GP, HB, and KH) needs to be listed in the material section. And the characteristics of early and late maturity should be introduced.
6. Although the authors obtained a unique marker by screening, there is no direct evidence whether this marker is actually associated with late maturation,
7. Although the authors obtained a unique marker through screening, there was no direct evidence whether this marker was really associated with late maturation, leading to the inaccurate title of the article, because the mutant has many different shapes from other varieties.
8. What gene do the unique markers come from? Where is the difference in the gene among varieties? Has the function of this gene been studied? The author does not mention or discuss
9. We suggest that the results be written separately from the discussion
Author Response
Response to reviewer 1
First of all, authors are grateful to reviewer 1 and editor for the valuable comments on article. Thank you for the time and consideration given in this regard. The response to reviewer 1 is as follows:
#1
We received proofreading from a native speaker working in a related field.
#2
We changed the title as below:
Development of late ripening citrus mutant, Ara-unshiu (Citrus unshiu) and its selectable marker
- In the first paragraph of the results section, the author wrote that the mutant was obtained by gamma irradiation treatment, but the treatment method and intensity were not mentioned in the materials and methods section.
#Response
We changed as below and added the references reported previously
To induce mutant lines of C. unshiu (Marc. cv. Miyagawa-wase), scions were irradiated with gamma rays (60Co, 100Gy with 60% survival rate of budsticks) followed by grafting on branches of Miyagawa-wase (C. unshiu Marc.) [31].
- The Figure 1 lacks the necessary scale, which mistakenly leads people to believe that the mutant is significantly smaller than WT, but the result is not as can be seen from Table 1.
#Response
We made a mistake the WT fruit photo as a other mutant fruit photo. The Fig.1 was changed as the correct WT fruit.
- What do the letters of a b L stand for in Table 2? Lack the necessary information.
#Response
We added the information in the Table 2
- The information of Table S5 is missing in the online document.
#Response
We added the information in the Table S5
- Citrus varieties (CG, GP, HB, and KH) needs to be listed in the material section. And the characteristics of early and late maturity should be introduced.
#Response
We added in the plant material section as below:
Other citrus varieties used in this study were as follows: Jedae-unshiu (J), Chung-Gyeon (CG), Gam-Pyong (GP), Hanla-Bong (HB), and Kara-Hyang (KH). Jedae-unshiu has an early maturity and the others have a late maturity trait.
- Although the authors obtained a unique marker by screening, there is no direct evidence whether this marker is actually associated with late maturation,
#Response
The selected marker (Ara-SNP4) is not linked with late maturation. It is a marker selected to distinguish with other citrus species for the intellectual property rights
- Although the authors obtained a unique marker through screening, there was no direct evidence whether this marker was really associated with late maturation, leading to the inaccurate title of the article, because the mutant has many different shapes from other varieties.
#Response
The selected marker (Ara-SNP4) is not linked with late maturation. It is a marker selected to distinguish with other citrus species for the intellectual property rights. And the SNP of Ara-SNP4 is located on the intergenic region of the genome.
- What gene do the unique markers come from? Where is the difference in the gene among varieties? Has the function of this gene been studied? The author does not mention or discuss
#Response
We added as below:
The SNP site of Ara-SNP4 marker was located on the intergenic region of the genome.
- We suggest that the results be written separately from the discussion
#Response
We submitted as a (short) communication. So, we would like to be combined the results with a discussion.

Reviewer 2 Report
The manuscript titled "Isolation, characterization, and identification of a specific selection marker for the Ara-unshiu citrus mutant with late fruit ripening" by Kim et al. reports the generation and characterization of a new citrus mutant, Ara-unshiu. It is an interesting variant that shows a unique fruit ripening phenotype.
Introduction:
- please provide some quantitative data on the economic importance of Citrus and put it briefly into perspective compared to other highly relevant types of fruit
- which molecular pathways influence the fruit ripening process? Are these known?
- What is the rationale behind naming the new citrus variety Ara-unshiu?
Methodology:
- please provide details on the exact parameters of the gamma ray treatment protocol (e.g., intensity, duration, used device, manufacturer, etc.)
- please provide oligonucleotide sequences (for the primers used in the PCRs)
- the PCR programs used for the DNA analysis (e.g., Fig. 3) should be provided
Results:
- Fig. 1: are the fruits shown to scale? The Ara-unshiu fruit look far smaller and overall inferior to the wt fruit. However, Table 1 indicates that Ara-unshiu produced overall larger and heavier fruit than the wt. Please clarify.
- Fig. 2: in my opinion it could be placed in the supplementary materials where it can be shown with larger lettering, the text of the figure is not readable.
- Fig. 3: PCR should always include a negative control (i. e., all components without template DNA) to rule out contamination with DNA
- Fig. 3: what are the expected sizes of the PCR products?
- Were the fruit of Ara-unshiu subjected to flavor testing? Testing the quality of the fruit in a blind taste test would be recommended.
Recommendation: a solid contribution that merits reconsideration for publication in Plants once my comments have been adequately addressed.
Author Response
Response to reviewer 2
First of all, authors are grateful to reviewer 2 and editor for the valuable comments on article. Thank you for the time and consideration given in this regard. The response to reviewer 2 is as follows:
Comments and Suggestions for Authors
The manuscript titled "Isolation, characterization, and identification of a specific selection marker for the Ara-unshiu citrus mutant with late fruit ripening" by Kim et al. reports the generation and characterization of a new citrus mutant, Ara-unshiu. It is an interesting variant that shows a unique fruit ripening phenotype.
# Response
We changed the title as below:
Development of late ripening citrus mutant, Ara-unshiu (Citrus unshiu) and its selectable marker
Introduction:
- please provide some quantitative data on the economic importance of Citrus and put it briefly into perspective compared to other highly relevant types of fruit
#Response
According to the 2023 Korea National Statistical Office report, Jeju Island's citrus cultiva-tion area is the largest at 15,818 ha, followed by plums and persimmons at 56 ha and 55 ha, respectively.
- which molecular pathways influence the fruit ripening process? Are these known?
#Response
We briefly mentioned in the introduction as below and the more detailed content was included in the transcriptome paper of Ara-unshiu preparing now.
“Fruit ripening programs are known to be mediated by the combined effects of plant hormones, transcription factors, and epigenetic modifications, all of which can affect fruit quality [16-18].”
- What is the rationale behind naming the new citrus variety Ara-unshiu?
#Response
Our university is located on Ara-Dong. That’s why we named as Ara-unshiu
Methodology:
- please provide details on the exact parameters of the gamma ray treatment protocol (e.g., intensity, duration, used device, manufacturer, etc.)
#Response
We changed as belows
To induce mutant lines of C. unshiu (Marc. cv. Miyagawa-wase), scions were irradiated with gamma rays (60Co, 100Gy with 60% survival rate of budsticks) followed by grafting on branches of Miyagawa-wase (C. unshiu Marc.) [31].
- please provide oligonucleotide sequences (for the primers used in the PCRs)
#Response
We added Table S5 (primer sequences)
- the PCR programs used for the DNA analysis (e.g., Fig. 3) should be provided
#Response
We mentioned on line 119~120
Results:
- Fig. 1: are the fruits shown to scale? The Ara-unshiu fruit look far smaller and overall inferior to the wt fruit. However, Table 1 indicates that Ara-unshiu produced overall larger and heavier fruit than the wt. Please clarify.
#Response
We made a mistake the WT fruit photo as a other mutant fruit photo. The Fig.1 was changed as the correct WT fruit.
- Fig. 2: in my opinion it could be placed in the supplementary materials where it can be shown with larger lettering, the text of the figure is not readable.
#Response
We changed the Fig. 2 as a scale-up size.
- Fig. 3: PCR should always include a negative control (i. e., all components without template DNA) to rule out contamination with DNA
#Response
We added the negative control in the Fig. 3.
- Fig. 3: what are the expected sizes of the PCR products?
#Response
We added the PCR product size.
- Were the fruit of Ara-unshiu subjected to flavor testing? Testing the quality of the fruit in a blind taste test would be recommended.
#Response
We didn’t the flavor test. Every year, we sampled and investigated the traits of our selected mutant lines on mature harvest periods. At that time we tested the taste of the harvested fruits from the several mutant lines. But we have not been the blind taste test. Keep in mind your recommendation.
Round 2
Reviewer 1 Report
The article has been improved a lot through revision and I have no other comments
Reviewer 2 Report
The authors addressed my previous comments adequately. As such, I support publication of the manuscript in the journal Plants.